# Performance Increases in Pair Skating and Ice Dance at International Championships and Olympic Games

**DOI:** 10.3390/ijerph191811806

**Published:** 2022-09-19

**Authors:** Thomas Rauer, Hans-Christoph Pape, Zoé Stehlin, Sandro Heining, Matthias Knobe, Tim Pohlemann, Bergita Ganse

**Affiliations:** 1Department of Trauma Surgery, University Hospital Zurich, 8091 Zurich, Switzerland; 2Faculty of Medicine, University of Zurich, 8032 Zurich, Switzerland; 3Department of Orthopedic and Trauma Surgery, Lucerne Cantonal Hospital, 6000 Luzern, Switzerland; 4Department of Trauma, Hand and Reconstructive Surgery, Saarland University, 66421 Homburg, Germany; 5Werner Siemens Foundation Endowed Chair of Innovative Implant Development, Saarland University, 66421 Homburg, Germany

**Keywords:** performance, figure skating, winter sports, elite athlete, trauma, age, competition

## Abstract

In pair skating and ice dance, performance seems to have increased at international competitions, which is potentially associated with changes in athlete age. We hypothesized increasing age, numbers of total points and more complex jumps of the best elite couples at international championships in recent years. Corresponding data were assessed via the results databases of the European and World Championships, as well as the Winter Olympics since 2005. Linear regression statistics were conducted, and significance was assessed via one-way ANOVAs. There were no significant changes in age. Increases in total points were found in both disciplines (World and European Championships both *p* < 0.001 for both disciplines, Olympics pair skating *p* = 0.003, ice dance n/a). Significant increases were found in the number of double and triple twist jumps at the European Championships (Double *p* = 0.046, triple *p* = 0.041), but not at the World Championships or the Olympics. At the World Championships, single solo jumps decreased (*p* = 0.031) in favor of triple jumps, which increased (*p* = 0.020), without a similar effect at the European Championships or Olympics. In conclusion, increases in total points and more complex jumps were observed at international championships without associated changes in age. Attention should be given to possible changes in the incidence of acute and overuse injuries following this development.

## 1. Introduction

Pair skating and ice dance at international level are among the five disciplines of figure skating, as defined by the International Skating Union (ISU) [1]. Pair skating and ice dance are both pair disciplines, but with different technical requirements and rules of their competition programs [2].

The difficulty of the programs seems to have increased in the pair disciplines [3], just as the number of turns in the jumps, as well as the average athlete age. A higher degree of complexity of the elements leads to a potentially higher score in both pair skating and ice dance. Due to the increasing difficulty of jumps with more rotations, the number of solo jumps, throw jumps and twist jumps might serve as an objective parameter of performance in pair skating, as could the total score in ice dance, and help to study performance trends. Performance trends have not been previously reported for pair skating and ice dance, but are of interest, e.g., to evaluate risk trends for injury and chronic strain. Since figure skating, unlike sports such as swimming, cycling or track and field [4,5,6,7], is not associated with results in times or distances, other measures need to be found to estimate the athletes’ performance for such an analysis. Therefore, in this study, to study performance in an objective way, the number of turns of the jumps were used as judged by a technical panel based on a restrictive set of technical rules. Falls and under-rotated or downgraded jumps were not considered. Only the definitive jump performance recognized by the technical jury was taken into account and evaluated. Of note, all athletes had the same amount of time for their warmup and performance, namely 4 min each in pair skating and in ice dance.

Due to the continuous development of figure skating with increasing technical and artistic demands, a high prevalence of sport-specific overuse injuries was described, including tendinitis, patellofemoral syndromes and lower back pain, as well as injuries, such as stress fractures and muscle strains [8,9,10,11,12]. Increased bone mineral density, trabecular bone mineral density and bone strength due to high impact forces were reported in the landing leg of figure skaters compared to the take-off leg [13,14]. In pairs skating, ice dance and synchronized skating, acute injuries predominate, while in singles skating, overuse injuries are more common than acute injuries [12]. Reasons for this discrepancy seem to include differences in the numbers of jumps and rotations, as well as risks associated with lifting elements in combination with skating in close proximity to the partner [15]. The acute and overuse injuries seem to have biomechanical causes and are associated with muscular imbalances in strength and flexibility, and inadequate core muscle strength [16]. Moreover, lower back pain is a known problem in figure skaters and was described in up to 13% of skaters [17]. The underlying reasons include the rigidity of the skates, that restrict knee and ankle movement and thus prevent adequate force absorption during jump landings [16]. In addition, certain compulsory elements of figure skating might contribute to the occurrence of lower back pain, such as various ring or Biellmann positions, which require extreme hyperextension of both, the lumbar spine and the hips to achieve a greater range of motion for advanced techniques [18]. Impact forces are higher in jumps with more rotations, that are thus connected to a greater risk of injury [14]. The reason for this is that the time it takes for the force to dissipate on landing is shorter when there are more turns [14,18]. Greater vertical velocities upon take-off were shown in very experienced athletes [18]. It has also been found that the risk of injury in figure skating correlates with age, and that older age is associated with more sports injuries [19].

We hypothesized increasing age and increasing numbers of total points and more complex jumps of the best couples who competed in pair skating and ice dance at the European and World Championships, and the Olympic Games in recent years.

## 2. Material and Methods

Age, total points and the numbers of single, double, triple and quadruple jumps of the best five couples, both in ice dance and pairs skating, were assessed via the results databases of the European and World Championships, as well as the Winter Olympics since 2005. The three championships were chosen to make sure that findings are generalizable. We chose to analyze the best five couples of each competition and group only, as the differences in performance are huge between the top couples and the lower-ranking ones.

The number of turns as well as the correct execution of the jumps are assessed during the championships by the technical panel. The technical panel consists of two technical specialists and a technical controller. Evaluation is conducted on the basis of a restrictive set of technical rules, so that an objective evaluation of the technical execution and the number of rotations performed is ensured. The following applies to our study: Falls, under-rotated or downgraded jumps were not considered. Only the definitive jump performance recognized by the technical jury was taken into account and evaluated.

In addition to the Total Points, we chose to analyse the number of rotations in the jumps as an objective measure of performance that reflects impact intensity and the forces acting on the athlete’s body in pair skating.

### Statistical Analyses

The datasets analyzed for this study are third party data and can be found on the website of the International Skating Union (ISU, access date 25 March 2022): https://www.isu.org/figure-skating/entries-results/fsk-results. Via this URL, others can access these data in the same manner as the authors. The authors did not have any special access privileges that others would not have.

IBM^®^ SPSS^®^ Statistics version 27 (Armonk, NJ, USA) was used for statistical analysis. Significance was assumed at *p* < 0.05. Age, total points and the numbers of single, double, triple and quadruple jumps were assessed via the results databases of the respective international championships as provided on the ISU website. For each year, data of the best five couples, both in ice dance and pairs skating, were pooled and analyzed for the European and World Championships, and the Olympic Games separately. Linear regression statistics were run to calculate the slope and the coefficient of determination for changes in age, total points and number of single, double, triple and quadruple jumps over time separately for the European and World Championships and the Olympic Games. Normal distributions of data were tested by the Kolmogorov–Smirnov and Shapiro–Wilk tests. Significance was assessed via one-way ANOVAs with the year as independent variable and the score, number of jumps or age as dependent variables.

## 3. Results

### 3.1. Age Trends

The average age was significantly lower in the women compared to the men (all championships pooled; pair skating: *p* < 0.001, women: 25.06 years +/− 1.83, men: 27.11 years +/− 1.58; ice dance, *p* < 0.001, women: 24.80 years +/− 2.15, men: 26.44 years +/− 1.61). There were no significant changes with time in the age of the best five athletes of each year for women or men in pair skating or ice dance (Pair skating: women slope 0.160, R^2^ = 0.247, *p* = 0.059; men slope −0.008, R^2^ = 0.001, *p* = 0.920; ice dance: women slope −0.239, R^2^ = 0.259, *p* = 0.052; men slope −0.129, R^2^ = 0.149, *p =* 0.155). In ice dance, a non-significant trend towards lower age was observed in women and men.

### 3.2. Total Points

Figure 1 shows trends of the average total points of the best five pairs in pair skating and ice dance at the European and World Figure Skating Championships, as well as the Olympic Games since 2005. Significant increases were found in both disciplines and in all three events. Significance could not be tested for ice dance at the Winter Olympics, as only two time points were available since the changes in regulations in 2011. However, the regression slope among the two events is comparable to the other Championships and therefore indicates a similar trend.

### 3.3. Numbers of Jumps

Figure 2 displays trends of the average number of jumps of the best five pairs in pair skating separated by solo jumps, throw jumps and twist jumps. The associated regression statistics are shown in Table 1. Significant increases were found in the double and triple twist jumps at the European Championships but not at the World Championships or the Olympics. At the World Championships, the number of single solo jumps decreased significantly in favor of triple jumps, which increased significantly, without a similar effect at the European Championships or Olympics.

## 4. Discussion

The present study was the first to analyze age and performance trends in pair skating and ice dance by assessing the average age and numbers of total points and jumps of the best couples who have competed at the European and World Championships, and Olympic Games in recent years. The results showed no significant changes in age. Significant increases in the total points were found in both disciplines and in all championships. In pair skating, significant increases were also found in the number of double and triple twist jumps at the European Championships, but not at the World Championships or the Olympics. At the World Championships, the number of single solo jumps significantly decreased in favor of triple jumps, without a similar effect in the European Championships or Olympics.

### 4.1. Age

We did not find any changes in age over time in neither pair skating nor ice dance. A constant age of international champions was also reported from swimming in World championships and Olympic Games [20]. What we did find, however, was a significant sex difference: women were on average around two years younger than men (pair skating and ice dance: 25.06 and 24.80 years in women compared to 27.11 and 26.44 years in men). It is well documented that the age of peak performance differs among sports and that it is higher the longer the event takes (longer in ultra-endurance events compared to e.g., sprint events) [21]. In comparison, in cross-country skiing, peak age was reported at 26.2 years in distance and 26.0 years in sprint events [22], just as in weightlifting (26 years), while elite power lifters had a peak age of 35 years [23]. In track and field, women on average had their peak age two years later than men (women 21.60 years and men 19.97 years), which is the opposite of what we found in the present study [7]. We do not have an explanation for this finding, but a possible reason is that younger female athletes are lighter and easier to throw in the throw jumps. Aerobic and anaerobic power are known to decline with age at a similar rate [24]. Physiologically, progressive losses of muscle mass, force- and power-generating capacity, flexibility and specific muscle tension characterize the age-related changes in the ageing athlete, while in the cardiovascular system, a declining cardiac output and stroke volume, as well as cardiac and vascular stiffness progressively decrease performance [25].

### 4.2. Performance Increases

We found increases in performance that are reflected in rising total point scores and jump complexity. Smaller performance increases were reported in sprint and distance running [26], while a plateauing of performance and in some cases even a decrease was observed in other sports [27,28,29,30,31]. When pole vault became Olympic for women, similarly fast increases in performance were observed as in the present study [32]. Speculatively, increases in total points are not due to more complex jumps, but potentially an adjustment to the recent changes in the regulations and the optimization of the programs in terms of scoring.

Far-reaching rule changes, such as the adjustment of the ice dance competitions from three to two competition parts with implementation in the 2011/2012 season, are reflected in a jump in the total point scores (Figure 1B). Especially in ice dance, the steady increase in total points since 2011 can most speculatively be explained by an adaptation of the ice dance teams to the recent changes in the regulations and the optimization of the programs in terms of scoring.

In addition, technical improvements in skate development, as well as innovations in training and preparation probably had a minor contribution to the increased scores. Custom-made skates with reduced weight, better stability support for the ankle joints and increasing comfort, as well as articulated figure skates were introduced with the aim to reduce impact forces and the incidence of injury [33]. Similarly, better ice and air quality of figure skating rinks might have improved conditions [34,35]. In addition, improvements have been implemented in training regimen. Progress was made in core and upper body strength and conditioning [18]. Off-skating training programs for figure skaters include training exercises that require eccentric and centric muscle actions and that allow adaptation to asymmetric or unilateral movements [18]. Strong abdominal and core muscles help to compensate for the forces encountered during jumps and landings, thus helping to prevent acute and overuse injuries [3,18].

### 4.3. Potential Effects on Injuries and Overuse Symptoms

Jumps with more rotations are known to be associated with impacts of greater magnitude and intensity [14,18,36]. Higher incidences of overuse injuries in the single disciplines were shown to be due to increasing technical difficulty with more difficult jumps and longer training times [12]. The increasing scores found in the present study should be monitored with regard to injury rates and chronic degenerative damage in athletes. Skaters are required to improve their physical skills such as agility, strength and power to achieve the increased level of technical difficulty without getting injured [18].

## 5. Conclusions

The present study was the first to analyze performance and age trends in elite pair skating and ice dance. We showed that at the European and World Championships, as well as at the Olympics, rising total points and in some instances more complex jumps are performed and suggested that further attention should be given to possible increases in the incidence of acute and overuse injuries after recent changes in regulations.

## Figures and Tables

**Figure 1 ijerph-19-11806-f001:**
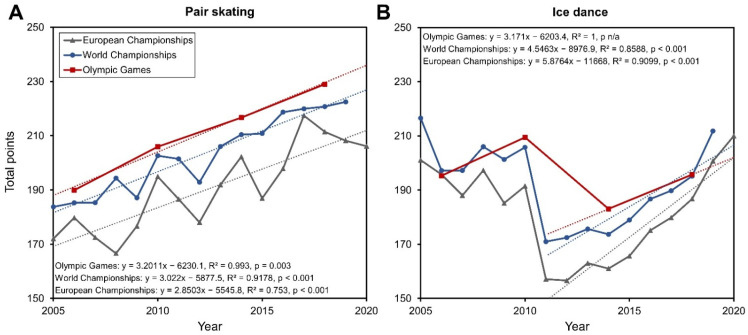
Trends of the average total points of the best five pairs in pair skating (**A**) and ice dance (**B**) at the European and World Figure Skating Championships, and the Olympic Games since 2005. Linear regression analysis was performed to show trends over time in pair skating (2005–2020) and ice dance (2011–2020). Regression functions, coefficients of determination (R^2^) and p values (ANOVA, dependent variable: average total points of the best five pairs, independent variable: year) are shown for each group. In ice dance, a drop in total points followed a change in regulations in 2010. Note the performance peaks in pair skating at the European Championships in years with Winter Olympics.

**Figure 2 ijerph-19-11806-f002:**
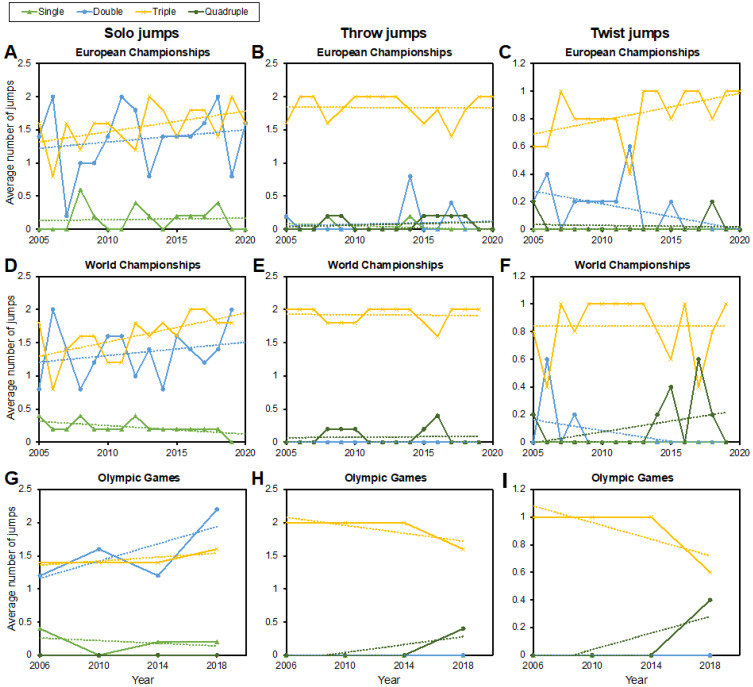
Jump types and trends of the average number of jumps of the best five pairs in pair skating. Results are separated by solo jumps (**A**,**D**,**G**), throw jumps (**B**,**E**,**H**) and twist jumps (**C**,**F**,**I**), and shown for the European (**A**–**C**) and World (**D**–**F**) Championships, as well as for the Olympic Games (**G**–**I**). See Table 1 for regression statistics.

**Table 1 ijerph-19-11806-t001:** Regression parameters for Figure 2 including slopes in jumps/year, coefficients of determination (R^2^) and the p values indicating if an increase or decrease was significant (significance highlighted in bold). Data are shown for pair skating. In ice dance, only jumps with a single rotation are allowed.

	Regression Parameters [Slope in Jumps/Year, R^2^, *p* Value]			
	Single	Double	Triple	Quadruple
European Championships				
Solo jumps	0.002, 0.004, 0.825	0.019, 0.031, 0.515	0.031, 0.473, 0.065	−0.019, 0.308, 0.246
Throw jumps	−0.006, 0.365, 0.165	0.004, 0.096, 0.723	−0.001, 0.000, 0.937	−0.006, 0.050, 0.404
Twist jumps	n/a	0.019, 0.255, **0.046**	0.012, 0.265, **0.041**	−0.001, 0.007, 0.763
World Championships				
Solo jumps	−0.013, 0.310, 0.031	0.020, 0.053, 0.410	0.044, 0.350, **0.020**	n/a
Throw jumps	n/a	n/a	−0.001, 0.003, 0.858	−0.001, 0.003, 0.858
Twist jumps	n/a	−0.016, 0.193, 0.101	0.000, 0.000, 1.000	0.016, 0.147, 0.158
Olympic Games				
Solo jumps	−0.01, 0.316, 0.684	0.065, 0.710, 0.290	0.015, 0.775, 0.225	n/a
Throw jumps	n/a	n/a	−0.03, 0.775, 0.225	0.03, 0.775, 0.225
Twist jumps	n/a	n/a	−0.03, 0.775, 0.225	0.03, 0.775, 0.225

## Data Availability

The datasets analyzed for this study are third party data and can be found on the website of the International Skating Union (ISU, access date 25 March 2022): https://www.isu.org/figure-skating/entries-results/fsk-results.

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
