# Peer review of "Performance Increases in Pair Skating and Ice Dance at International Championships and Olympic Games"

_ijerph, 2022, doi:10.3390/ijerph191811806_

Round 1
Reviewer 1 Report
General:
I would like to commend the authors for their efforts in undertaking this research in an under-researched population, and where methods of assessing injury risk etc. are needed given the injury profiles within the sport. However, having reviewed the manuscript, I am (unfortunately) providing the recommendation of a rejection. This decision is based upon a number of factors that I hope the authors take in to consideration:
1. I feel the manuscript title does not match the manuscript contents – the title suggests to highlight implications for injury/overuse, however no such analysis has been conducted and is only a 3-4 sentence section within the discussion
2. The aim/hypothesis is not linked to injury/overuse (line 80 – 83) therefore the nature of the study/rationale needs revising
3. I think the manuscript is not assisted by the fact that no injury audit data has been collected (likely due to methods adopted) and subsequently not included within analysis, meaning that the analysis conducted, and any link to injury/overuse is speculative.
4. Whilst I appreciate a greater degree of complexity leads to a potential greater points accumulation, this has not been highlighted in the introduction/rationale in enough depth. Subsequently, I fail to see how points accumulation may be linked to injury/overuse as per the manuscript/title. Similarly, for age, I accept the rationale/citations re: increased age = potential increase injuries, however injury data is not present, therefore this form of analysis, is (again) speculative
5. The discussion lacks some critical analysis of variables mentioned at line 280 - 282, as a result, the link to injury/overuse or the proposed mechanisms to combat these is lacking
I have provided further details of relevant sections of the manuscript below:
Abstract:
Line 17: You state that a high prevalence of injury/overuse injury was observed in ice dance/figure skating, yet terminology used in introduction (please see below) references (and citations) relate to figure skating – consistency of terminology needed
Line 18: As highlighted in general comments, there is no aim/hypothesis related to injury/overuse, meaning the study lacks some clarity
Introduction:
Line 37: The terminology has changed here to figure skating – the title, and opening sentences of the introduction refer to ice dance and pair skating. Whilst this change is likely related to the references used, consistency of terminology is needed here, and throughout the manuscript.
Line 37 – 77: I have some concerns regarding the information provided here and how these link to the aims/methods of the paper. Whilst the information and supplied references are appropriate, I feel the information does not match a) the aim of the paper (no mention of injury/overuse within the aim/hypothesis) or b) provide a rationale for the methods adopted whereby the authors have looked at results of competitions/numbers of jumps etc.
Line 75 – 77: I feel this line would be better suited within the methods section
Regarding your aim/hypothesis what is the relevance of the comparisons between championships? How does this link to injury/overuse? This is unclear and needs consideration here.
Materials & Methods:
I feel that large portions (namely line 88 – 155) of this section can be removed (or as a minimum condensed down to a few sentences to cover all aspects) as they are not relevant to the specific methods adopted by the authors within the present study and bear no relevance if the study were to be replicated. For example, Sections 2.1 (Competition Rules), 2.2 (Disciplines and Their Specifications) and 2.3. (Definition of Specific Pair Skating Elements) are generic explanations of the rules/regulations of the relevant sports – not methods. In reading the methods, a researcher would be able to replicate the study based upon the information provided in line 84 – 87 and 156 – 173 (Section 2.4; Statistical Analysis) of the current manuscript – therefore I would suggest the authors consider revising this section.
Line 167: What type of regression statistics? Can the authors be more explicit here please.
Results:
As highlighted in general comments – I feel the rationale in introduction needs to be strengthen to consider age/total points as factors relating to injury/overuse. Particularly age – this is based upon (from a coaching perspective) as you become more proficient as you get older, skill sets increase, therefore more jumps/technically difficult components are used more frequently?
As highlighted in introduction section, what is the relevance of the comparisons between championships? How does this link to injury/overuse? This is unclear and needs consideration within the introduction/rationale to consider these results.
Table 1: The information/formatting provided in this table may need to be considered as it is difficult to follow and appears quite condensed.
Discussion:
Given the points raised throughout, I feel this section needs some consideration as to the points discussed here. There is minimal (3-4 sentences) of talking points relating to injury/overuse (line 275 – 282 of current manuscript), when this is the supposed focus of the paper.
Age (Section 4.1): No age differences detected, however you then discuss sex/gender differences – this is potentially an interesting result, however the subsequent discussion becomes disjointed as the supposed focus of the paper is relating to injury/overuse, however age of peak performance is subsequently discussed against sports that (could be considered) not comparable to ice dance/pairs skating (e.g. powerlifting etc.) – I feel here the focus needs to be on potential mechanisms of injury/overuse with this difference. For example, menstrual cycle/ACL laxity and bone formation mechanisms are a potential avenues that could be discussed.
Line 243 – 248: How do these findings link directly to your results? You have identified no differences in age, yet then discuss some physiological parameters that (appear to) contradict your finings, yet this is not mentioned
Performance Increases (Section 4.2): I am unsure as to the title of this section as the authors have not tested ‘performance’ – total points and number of jumps have been analysed. It is well-established that increased jump complexity = greater potential for points (assuming executed correctly), therefore I do not see the novelty in this statement (line 250). Similarly, the analysis and comparison between some of these sports is confusing – pole vault vs. ice dance?
Line 254 – 256: This is a speculative statement as the analysis of scoring systems has not been analysed per se, therefore I feel the authors need to change this statement to reflect the speculative nature (i.e. ‘Speculatively, a potential change of the increase in total points, is not in more complex jumps, but potentially an adjustment to the recent changes in the regulations and the optimisation of the programmes in terms of scoring.’). Similarly, using the term ‘massive increase’ is colloquial language, please amend to reflect a more scientific approach (I have removed from my proposed revised sentence).
Line 263 – 266: Similar to above points, the speculative nature of these variables needs to be highlighted more explicitly by the authors here, especially as these were not considered in the design/analysis of the present study. No issue with these being included, however, as I say, need to highlight the speculative nature.
Line 267 – 268: Similarly, the statement re: air quality etc. is highly speculative. Whilst the authors have cited x2 references to support this point, there is no link from the authors to the variables within these studies ( and their study aims (i.e. injury/overuse) therefore I do not feel this is relevant. One of the studies cited here has reviewed data from 1971 – 1989 in ice arenas, some 15-16 years before the analysis of the championships included within the present study.
Line 272 – 274: References needed to support this statement
Implications for Injuries and Overuse Symptoms (Section 4.3): Given this is the proposed area of the manuscript, I was disappointed to see this section is 3-4 sentences long. At line 280 – 282 the authors cite a few variables that I feel should be the foundation of the discussion section and how these variables may be influenced by the variables they have analysed.
Author Response
We are thankful for the useful comments by the reviewers that have helped to strengthen our manuscript! Our responses are highlighted in red, and changes in the manuscript are highlighted using ‚track changes‘. We hope the reviewers will find our manuscript acceptable for publication in its current state.
Reviewer 1:
General:
I would like to commend the authors for their efforts in undertaking this research in an under-researched population, and where methods of assessing injury risk etc. are needed given the injury profiles within the sport. However, having reviewed the manuscript, I am (unfortunately) providing the recommendation of a rejection. This decision is based upon a number of factors that I hope the authors take into consideration:
- I feel the manuscript title does not match the manuscript contents – the title suggests to highlight implications for injury/overuse, however no such analysis has been conducted and is only a 3-4 sentence section within the discussion
Response: We are thankful for the reviewer’s comment. As the title was indeed misleading, the addition of "Implications for Acute and Overuse Injuries" was deleted. The main focus of our study is and was the performance and age development in pair skating and ice dance since 2005 at international championships and Olympic Games.
- The aim/hypothesis is not linked to injury/overuse (line 80 – 83) therefore the nature of the study/rationale needs revising
Response: With the adaptation of the title, the focus of our study, namely the evaluation of the performance development in pair skating and ice dance since 2005, is clearly emphasized and thus the hypothesis of our study now becomes conclusive. Therefore, no adjustment of the hypothesis was made in the manuscript.
- I think the manuscript is not assisted by the fact that no injury audit data has been collected (likely due to methods adopted) and subsequently not included within analysis, meaning that the analysis conducted, and any link to injury/overuse is speculative.
Response: The authors are thankful for the reviewer’s comment and have adapted the manuscript thoroughly. Data on injury were neither recorded, nor are they available to be added to the current analysis. As trauma surgeons, our focus is on injuries, that are likely to occur more frequently when the athletes’ feet and ankles are subjected to higher forces.
- Whilst I appreciate a greater degree of complexity leads to a potential greater points accumulation, this has not been highlighted in the introduction/rationale in enough depth. Subsequently, I fail to see how points accumulation may be linked to injury/overuse as per the manuscript/title. Similarly, for age, I accept the rationale/citations re: increased age = potential increase injuries, however injury data is not present, therefore this form of analysis, is (again) speculative.
Response: We have added the following sentence to the introduction: “A higher degree of complexity of the elements leads to a potentially higher score in both pair skating and ice dance.”
- The discussion lacks some critical analysis of variables mentioned at line 280 - 282, as a result, the link to injury/overuse or the proposed mechanisms to combat these is lacking.
Response: The title of subsection 4.3 in the discussion section has been changed to "Potential Effects on Injuries and Overuse Symptoms" so that this subsection addresses potential effects of performance development on injuries and overuse injuries. The critical discussion of the main focus, performance development, and age development in pair skating and ice dance was adequately discussed previously.
I have provided further details of relevant sections of the manuscript below:
Abstract:
Line 17: You state that a high prevalence of injury/overuse injury was observed in ice dance/figure skating, yet terminology used in introduction (please see below) references (and citations) relate to figure skating – consistency of terminology needed
Response: Pair skating and ice dance are among the five disciplines of figure skating as an international competitive sport. This is stated in the introduction. Further, the injury profile of figure skating as a whole and the differences in the individual disciplines are described here. Thus, there is a logical consistency of terminology here, which results from the context.
Line 18: As highlighted in general comments, there is no aim/hypothesis related to injury/overuse, meaning the study lacks some clarity
Response: See previous responses.
Introduction:
Line 37: The terminology has changed here to figure skating – the title, and opening sentences of the introduction refer to ice dance and pair skating. Whilst this change is likely related to the references used, consistency of terminology is needed here, and throughout the manuscript.
Response: See above response regarding consistency of terminology.
Line 37 – 77: I have some concerns regarding the information provided here and how these link to the aims/methods of the paper. Whilst the information and supplied references are appropriate, I feel the information does not match a) the aim of the paper (no mention of injury/overuse within the aim/hypothesis) or b) provide a rationale for the methods adopted whereby the authors have looked at results of competitions/numbers of jumps etc.
Response: By adjusting the title and hereby focusing on performance and age development, the logical connection between the hypothesis, the data collected, and the data collection method was established.
Line 75 – 77: I feel this line would be better suited within the methods section
Response: The mentioned two sentences were deleted in the introduction. In the methods section, the following was added: “The following applies to our study: Falls, under-rotated or downgraded jumps were not considered. Only the definitive jump performance recognized by the technical jury was taken into account and evaluated.”
Regarding your aim/hypothesis what is the relevance of the comparisons between championships? How does this link to injury/overuse? This is unclear and needs consideration here.
Response: We are thankful for this comment. The comparison between championships strengthens the argument that increases are a consistent finding that is not limited to only one championship. We have added this information to the introduction to make it clear to the readers why it is of importance: ‘The three championships were chosen to make sure that findings are generalizable.’
Materials & Methods:
I feel that large portions (namely line 88 – 155) of this section can be removed (or as a minimum condensed down to a few sentences to cover all aspects) as they are not relevant to the specific methods adopted by the authors within the present study and bear no relevance if the study were to be replicated. For example, Sections 2.1 (Competition Rules), 2.2 (Disciplines and Their Specifications) and 2.3. (Definition of Specific Pair Skating Elements) are generic explanations of the rules/regulations of the relevant sports – not methods. In reading the methods, a researcher would be able to replicate the study based upon the information provided in line 84 – 87 and 156 – 173 (Section 2.4; Statistical Analysis) of the current manuscript – therefore I would suggest the authors consider revising this section.
Response: In accordance with the reviewer's recommendations, we have shortened the methods section. Sections 2.1 (Competition Rules), 2.2 (Disciplines and Their Specifications) and 2.3. (Definition of Specific Pair Skating Elements) have been deleted and significantly condensed. Due to the complexity of the rules and regulations in this sport, we still think that readers who are not familiar with the details of the sport would have appreciated these sections, but we have done as the reviewer suggested.
Line 167: What type of regression statistics? Can the authors be more explicit here please.
Response: Linear regression was conducted. We have added this information to the statistics section of Materials and Methods.
Results:
As highlighted in general comments – I feel the rationale in introduction needs to be strengthen to consider age/total points as factors relating to injury/overuse. Particularly age – this is based upon (from a coaching perspective) as you become more proficient as you get older, skill sets increase, therefore more jumps/technically difficult components are used more frequently?
As highlighted in introduction section, what is the relevance of the comparisons between championships? How does this link to injury/overuse? This is unclear and needs consideration within the introduction/rationale to consider these results.
Response: With the adjustment of the title and the clarification of the hypothesis, the relevance of the injury/overuse argument is superfluous.
Table 1: The information/formatting provided in this table may need to be considered as it is difficult to follow and appears quite condensed.
Response: We have purposely provided the information in a condensed way and still think that this is a very adequate table.
Discussion:
Given the points raised throughout, I feel this section needs some consideration as to the points discussed here. There is minimal (3-4 sentences) of talking points relating to injury/overuse (line 275 – 282 of current manuscript), when this is the supposed focus of the paper.
Response: After adapting the initially misleading title and study focus, the discussion now seems appropriate.
Age (Section 4.1): No age differences detected, however you then discuss sex/gender differences – this is potentially an interesting result, however the subsequent discussion becomes disjointed as the supposed focus of the paper is relating to injury/overuse, however age of peak performance is subsequently discussed against sports that (could be considered) not comparable to ice dance/pairs skating (e.g. powerlifting etc.) – I feel here the focus needs to be on potential mechanisms of injury/overuse with this difference. For example, menstrual cycle/ACL laxity and bone formation mechanisms are a potential avenues that could be discussed.
Response: We have made some changes to improve the structure. After adapting the initially misleading title, the focus of our study, the evaluation of performance and age development in pair skating and ice dance, is now clear. The critical discussion of the main focus, performance and age development in pair skating and ice dance now seems of adequate depth. Since the survey of injuries and overuse injuries is not the focus of this study, the mentioned subsection now seems appropriate.
Line 243 – 248: How do these findings link directly to your results? You have identified no differences in age, yet then discuss some physiological parameters that (appear to) contradict your finings, yet this is not mentioned
Response: As we have found age differences between sexes, we think it is worth discussing these parameters.
Performance Increases (Section 4.2): I am unsure as to the title of this section as the authors have not tested ‘performance’ – total points and number of jumps have been analysed. It is well-established that increased jump complexity = greater potential for points (assuming executed correctly), therefore I do not see the novelty in this statement (line 250). Similarly, the analysis and comparison between some of these sports is confusing – pole vault vs. ice dance?
Response: In the introduction, the authors have thoroughly explained why it is hard to measure performance directly (and never has), and why the chosen parameters are considered the best way to indirectly assess performance in a sport that does not deliver results in meters or times. As stated in the beginning of the discussion, the present study is the first to analyse age and performance trends in pair skating and ice dance by assessing the average age and numbers of total points and jumps of the best couples. The specificity of the results referred to in the statement in line 250 results from this.
Line 254 – 256: This is a speculative statement as the analysis of scoring systems has not been analysed per se, therefore I feel the authors need to change this statement to reflect the speculative nature (i.e. ‘Speculatively, a potential change of the increase in total points, is not in more complex jumps, but potentially an adjustment to the recent changes in the regulations and the optimisation of the programmes in terms of scoring.’). Similarly, using the term ‘massive increase’ is colloquial language, please amend to reflect a more scientific approach (I have removed from my proposed revised sentence).
Response: The sentence was adjusted accordingly and the reviewer's suggestion was adopted.
Line 263 – 266: Similar to above points, the speculative nature of these variables needs to be highlighted more explicitly by the authors here, especially as these were not considered in the design/analysis of the present study. No issue with these being included, however, as I say, need to highlight the speculative nature.
Response: The sentence was adjusted as follows: “Especially in ice dance, the steady increase in total points since 2011 can most likely speculatively be explained by an adaptation of the ice dance teams to the recent changes in the regulations and the optimization of the programmes in terms of scoring.”
Line 267 – 268: Similarly, the statement re: air quality etc. is highly speculative. Whilst the authors have cited x2 references to support this point, there is no link from the authors to the variables within these studies ( and their study aims (i.e. injury/overuse) therefore I do not feel this is relevant. One of the studies cited here has reviewed data from 1971 – 1989 in ice arenas, some 15-16 years before the analysis of the championships included within the present study.
Response: The wording in this section is already in the subjunctive, which is grammatically a speculative form of possibility. Regarding the ice and air quality in the ice rinks, there are no more recent data. The injury issue should now be resolved.
Line 272 – 274: References needed to support this statement
Response: Done.
Implications for Injuries and Overuse Symptoms (Section 4.3): Given this is the proposed area of the manuscript, I was disappointed to see this section is 3-4 sentences long. At line 280 – 282 the authors cite a few variables that I feel should be the foundation of the discussion section and how these variables may be influenced by the variables they have analysed.
Response: The critical issues of the manuscript, performance and age development in pair skating and ice dance were adequately discussed. Since the survey of injuries and overuse injuries is not the focus of this study, the mentioned subsection in the discussion section seems adequate.
Reviewer 2 Report
Comments to the Author
This paper has the potential to be of interest to the readership, but needs major revision to be considered for publication.
The biggest problem in this study is that nothing about injury has been investigated. This study only examines and discusses performance skating and dance. Based on the results of this study, nothing is clear about the relationship between performance and acute and overuse injuries.
Title
The following text needs to be removed. Because you have not done any research about implications for injuries.
“-Implications for Acute and Overuse Injuries”
Introduction
In the first paragraph, you describe a lot about the correlation between skating and injury. However, this study does not examine anything about injuries. You need to make a major revision. The relationship between injury and performance should be discussed in the discussion.
Materials and Methods
Study design should be mentioned.
・How many players are included?
・A description of each athlete's demography is required(nationality, athletic history, past injury history…etc).
・Ethics should be described.
Results
・Are there any injured players?
・You need to indicate the total time of the competitions of each athletes in each competition
Discussion
・Why were women on average around two years younger than men in pair skating and ice dancing? No discussion of the reasons for this is given.
Conclusions
Possible increases in the inincidence of acute and overuse injuries are not suggested in this study.
Author Response
We are thankful for the useful comments by the reviewers that have helped to strengthen our manuscript! Our responses are highlighted in red, and changes in the manuscript are highlighted using ‚track changes‘. We hope the reviewers will find our manuscript acceptable for publication in its current state.
Reviewer 2:
Comments to the Author
This paper has the potential to be of interest to the readership, but needs major revision to be considered for publication.
The biggest problem in this study is that nothing about injury has been investigated. This study only examines and discusses performance skating and dance. Based on the results of this study, nothing is clear about the relationship between performance and acute and overuse injuries.
Title
The following text needs to be removed. Because you have not done any research about implications for injuries.
“-Implications for Acute and Overuse Injuries”
Response: We have done so. The title was misleading, so the addition of "Implications for Acute and Overuse Injuries" was deleted.
Introduction
In the first paragraph, you describe a lot about the correlation between skating and injury. However, this study does not examine anything about injuries. You need to make a major revision. The relationship between injury and performance should be discussed in the discussion.
Response: We have re-written the introduction and feel that it now matches the study aims and data.
Materials and Methods
Study design should be mentioned.
・How many players are included?
Response: The Material and Methods section has been adjusted as follows: “Age, total points and the numbers of single, double, triple and quadruple jumps of the best five couples, both in ice dance and pairs skating, were assessed via the results databases of the European and World Championships, as well as the Winter Olympics since 2005. We chose to analyze the best five couples of each competition and group only, as the differences in performance are huge between the top couples and the lower-ranking ones.”
Furthermore, the statistics section was also adjusted as follows: “For each year, data of the best five couples, both in ice dance and pairs skating, were pooled and analysed for the European and World Championships, and the Olympic Games separately.”
・A description of each athlete's demography is required (nationality, athletic history, past injury history…etc).
Response: Since the focus of the manuscript was changed away from injury, we do not consider this adequate anymore.
・Ethics should be described.
Response: In accordance with the journal regulations, the ethical information can be found at the end of the manuscript as follows:
Institutional Review Board Statement: The study was conducted in accordance with the Declaration of Helsinki, and approved by the Institutional Review Board of Saarland Medical Board (Ärztekammer des Saarlandes, application number 135/21).
Results
・Are there any injured players?
Response: By adapting the initially misleading title, the focus of our study was shifted to the evaluation of performance and age development in pair skating and ice dance.
・You need to indicate the total time of the competitions of each athletes in each competition
Response: All athletes have the same amount of time for their warmup and performance, namely 4 minutes each in pair skating and in ice dance. We have added this information to the introduction.
Discussion
・Why were women on average around two years younger than men in pair skating and ice dancing? No discussion of the reasons for this is given.
Response: We have added a discussion on athlete age to the discussion section and speculate that younger female athletes are lighter and easier to throw in the throw jumps.
Conclusions
Possible increases in the incidence of acute and overuse injuries are not suggested in this study.
Response: We have changed the focus of the manuscript and discussion accordingly.
Round 2
Reviewer 2 Report
The manuscript is well revised.